# Sensitivity of *Hydra vulgaris* to Nanosilver for Environmental Applications

**DOI:** 10.3390/toxics10110695

**Published:** 2022-11-17

**Authors:** Arianna Bellingeri, Chiara Battocchio, Claudia Faleri, Giuseppe Protano, Iole Venditti, Ilaria Corsi

**Affiliations:** 1Department of Physical, Earth and Environmental Sciences, University of Siena, 53100 Siena, Italy; 2Department of Sciences, Roma Tre University of Rome, 00146 Rome, Italy; 3Department of Life Sciences, University of Siena, 53100 Siena, Italy

**Keywords:** nanosilver, ecotoxicity, cnidarian, *Hydra*, sensing, water treatment

## Abstract

Nanosilver applications, including sensing and water treatment, have significantly increased in recent years, although safety for humans and the environment is still under debate. Here, we tested the environmental safety of a novel formulation of silver nanoparticles functionalized with citrate and L-cysteine (AgNPcitLcys) on freshwater cnidarian *Hydra vulgaris* as an emerging ecotoxicological model for the safety of engineered nanomaterials. AgNPcitLcys behavior was characterized by dynamic light scattering (DLS), while Ag release was measured by inductively coupled plasma mass spectrometry (ICP-MS). *H. vulgaris* (*n* = 12) subjects were evaluated for morphological aberration after 96 h of exposure and regeneration ability after 96 h and 7 days of exposure, after which the predatory ability was also assessed. The results show a low dissolution of AgNPcitLcys in *Hydra* medium (max 0.146% of nominal AgNPcitLcys concentration) and highlight a lack of ecotoxicological effects, both on morphology and regeneration, confirming the protective role of the double coating against AgNP biological effects. Predatory ability evaluation suggests a mild impairment of the entangling capacity or of the functionality of the tentacles, as the number of preys killed but not ingested was higher than the controls in all exposed animals. While their long-term sub-lethal effects still need to be further evaluated on *H. vulgaris*, AgNPcitLcys appears to be a promising tool for environmental applications, for instance, for water treatment and sensing.

## 1. Introduction

Silver nanoparticles (AgNPs) are widely employed in consumer products, especially for their efficient antimicrobial properties [1]. Their fields of application are growing, and thanks to their unique properties, AgNPs can be used for many different purposes, such as in electronic devices, as anticancer agents, and for water pollution sensing and remediation [2,3]. The environmental application of AgNPs is promising in terms of sensitivity to selected substances but needs careful consideration of the potential risks associated with AgNP release into the environment and the possible release of hazardous Ag ions [4]. The selection of the appropriate coating plays a fundamental role, both in defining AgNP sensitivity to certain substances and in controlling the release of toxic Ag ions [2,5]. Coatings containing reduced sulfur groups, such as thiols, have been demonstrated to successfully reduce the degree of dissolution of AgNPs, thereby lowering their (eco)toxicity [6,7,8,9].

Although having lower genetic complexity, aquatic invertebrate species with adult stem cells (ASCs) have been increasingly promoted in ecotoxicology, as they represent a valuable and reliable tool for understanding fundamental biological processes, the mode of action (MoA) of pollutants, and mechanisms of epigenetic toxicity [10]. *Hydra* sp. has been recently promoted as a suitable model for ecological risk assessment of legacy and contaminants of emerging concerns (CECs), including engineered nanomaterials (ENMs) [11,12]. They have been considered suitable as a bioindicator for freshwater ecosystems due to having a simple body organization which determines a high level of interaction between cells and the surrounding environment, facilitating permeation of toxic substances and uptake of ENMs [13,14]. *Hydra* sp. organisms are known for their longevity and ability to regenerate their bodies due to the life-long presence of ASCs [15]. They have been widely used in ENM ecotoxicity assessment at the individual level by investigating changes in morphology (shorter and/or clubbed tentacles or even disappearance), feeding (unable to catch prey), budding (unable to renew cells), and transcriptomics [16,17,18,19,20,21]. *Hydra* sp. has also been recently used to assess the safety of ENMs for environmental and biomedical applications, including AgNPs [4,22,23,24]. Interestingly, recent findings show that extracts from *Hydra* sp. basal discs, which contain antimicrobial peptides (AMP) and are involved in innate immune defense, reduce the toxicity of polyvinylpyrrolidone (PVP)-coated AgNPs to bacteria and zebrafish [25,26].

In the present study, the environmental safety of hydrophilic silver nanoparticles (AgNPs) bifunctionalized with citrate (cit) and L-cysteine (Lcys), was analyzed using the freshwater polyp *Hydra vulgaris*. This work is a follow up of our previous study in which the safety of the same batch of AgNPcitLcys was tested on microalgae and microcrustaceans belonging to freshwater and marine environments upon acute and chronic exposure [9]. The aim of this study was two-fold: firstly, we wanted to deepen the current knowledge of the environmental safety of AgNPcitLcys by adding a new trophic level to those already investigated and for which low/no toxicity was observed upon acute exposure (i.e., microalgae and microcrustaceans). AgNPcitLcys particles were synthesized for environmental applications (for both monitoring and remediation of contaminated waters) as they showed sensitivity to Hg, a known water pollutant. The assessment of AgNPcitLcys environmental safety is, hence, a priority for promoting their safe application by limiting any potential risk towards aquatic organisms. In addition, based on the available literature data on AgNP ecotoxicity and considering their widespread use in consumer products and many other applications including environmental monitoring and remediation [2,3], it is now necessary to further investigate biological targets of the aquatic environment, including new promising ones such as *Hydra*. This will prevent any unexpected ecotoxicity and support strategies for a more comprehensive eco-design of engineered nanomaterials for environmental applications [27].

## 2. Materials and Methods

### 2.1. Nanosilver Characterization

Silver nanoparticles with citrate and L-cysteine capping (AgNPcitLcys) were synthetized by the Department of Sciences, Roma Tre University of Rome, according to the procedure already described in Prosposito et al. [8] and Bellingeri et al. [9]. Briefly, 10 mL of water solution of sodium citrate (0.01 M), 25 mL of water solution of L-Cys (0.002 M), and 2.5 mL of water solution of silver nitrate (0.05 M) were mixed and degassed with Argon for 10 min. Then, 4 mL of water solution of NaBH_4_ (4 mg/mL) was added to the mix, and after 2 h, the product was recovered and purified by centrifugation (13,000 rpm, 10 min, 2 times with deionized water). The AgNPcitLcys particles, characterized by UV-Vis measurements in distilled water (by using Shimadzu 2401 PC UV-Vis spectrophotometer), showed the typical plasmonic peak at 420 nm due their nanosize. AgNPscitLcys particles were also investigated by synchrotron radiation-induced X-ray photoelectron spectroscopy (SR-XPS), confirming the molecular structure reproducibility and stability towards aging of the nanostructured material [8,28].

AgNPcitLcys aqueous suspensions (50 mg/L) were characterized by Dynamic Light Scattering (DLS) (Zetasizer Nano ZS90, Malvern, combined with the Zetasizer Nano Series software, version 7.02, Particular Sciences, Dublin, Ireland). Hydrodynamic diameter (HDd, nm), polydispersity index (PDI), and surface charge (ζ-potential, mV) were measured in MilliQ water and Hydra medium at 25 °C. The release of Ag from the AgNPcitLcys was assessed at the beginning (1 h) and at the end of the exposure times (96 h and 7 days) in *Hydra* medium. Ag release was measured at the highest AgNPcitLcys concentration tested (1000 μg/L). The solution was kept under the same conditions of temperature, light, and photoperiod as used for toxicity tests (18 °C, 400 lux, and 16/8 h light/dark photoperiod) and was mixed by manual shaking once a day. An aliquot was taken at each measured time point (1 h, 96 h, and 7 days) and centrifuged (5000× *g*, 40 min) by using a centrifugal filter device with a 3 kDa cut off (Amicon Ultra-15 mL, Millipore, Burlington, MA, USA). The resulting filtrate was acidified with HNO_3_ (10%) and analyzed for Ag concentration. As a control, Ag concentrations were also measured in *Hydra* medium with no addition of AgNPcitLcys. Ag concentrations were determined by inductively coupled plasma-mass spectrometry (ICP-MS) using the Perkin Elmer NexION 350 spectrometer (Waltham, MA, USA). The analytical accuracy was checked by comparing the certified and measured Ag concentration in the standard reference material SRM 1643e (Trace Elements in Water) of the National Institute of Standards and Technology (NIST). The analytical precision was evaluated by means of the percentage relative standard deviation (% RSD) of five replicate analyses of each water sample. The dissolution percentages were calculated as follows: (Ag_solution_ × 100)/AgNPcitLcys_concentration_.

### 2.2. Acute Short-Term Toxicity

The freshwater cnidarian *H. vulgaris* subjects, kindly provided by Prof. Massimilano Scalici of the Department of Sciences of Roma Tre University of Rome (Italy), were kept under controlled conditions in the laboratory in freshly prepared *Hydra* medium at 18 °C with a 16/8 h light/dark photoperiod at 400 lux. *Hydra* medium was prepared by the addition of 1 mL of CaCl_2_2H_2_O (1 M) and 1 mL of NaHCO_3_ (1 M) to a final volume of 1 L of MilliQ water (MilliQ). The polyps were fed ad libitum twice a week with newly hatched brine shrimp *Artemia franciscana* larvae, which were rinsed in MilliQ before being administered to the polyps. Individuals were not fed 4 days prior to the beginning of the test.

Ecotoxicity assays were performed in 24-well polystyrene plates with two polyps in each well in a final volume of 3 mL. Each concentration was performed in 6 replicates, with a total of 12 specimens for each tested concentration. The polyps were exposed to AgNPcitLcys and AgNO_3_ (as reference) at 0 (CTRL), 10, 100, and 1000 µg/L and 0 (CTRL), 10, 50, and 100 µg Ag/L, respectively. During the test, we ensured that pH values of exposure solutions stayed in the physiological range of 7.8–8, as that of *Hydra* medium with no AgNPcitLcys. This was facilitated by the fact that AgNPcitLcys were suspended in water with no added surfactants or additives and that the volume of AgNPcitLcys stock solutions added to each exposure well did not exceed 20 µL.

Two ecotoxicity tests were performed: (i) acute exposure at 96 h with morphological assessment of exposed polyps; (ii) acute exposure at 96 h with assessment of regeneration ability according to Wilby [29]. The regeneration assay was prolonged for up to 7 days in order to evaluate a sub-chronic exposure scenario, after which the predation ability of regenerated polyps was evaluated. This test was performed by feeding 1 individual with 10 *A. franciscana* larvae for 45 min in 10 mL of fresh *Hydra* medium in a 6-well polystyrene plate. A total of 6 polyps for each exposure concentration were employed for the evaluation of predation ability. At the end of the 45 min, the polyps were transferred and the number of eaten *A. franciscana* larvae was registered as well as that of dead *A. franciscana* larvae found at the bottom of the well.

## 3. Results and Discussion

### 3.1. AgNPcitLys Behavior in Hydra Medium

The hydrodynamic diameter (HD_d_) (Table 1) of AgNPcitLcys in MilliQ is consistent with the previous measurements reported in Prosposito et al. [8] and Bellingeri et al. [9], with a mean value of 136 ± 11 nm as measured by intensity and a net negative surface charge (–47.9 mV), as expected based on the surface functionalization with citrate and L-cysteine. The *Hydra* medium caused a moderate aggregation of AgNPcitLcys with an HD_d_ of 676 ± 10 nm, and an increase in ζ-potential to –18 mV. The presence of salts composing the *Hydra* medium probably caused a partial shielding of the negative charges of AgNPcitLcys, thereby reducing the electrostatic repulsion and leading to the formation of aggregates. The analysis of Ag release (Table 2) confirmed the low dissolution behavior previously reported for this particular kind of double-coated AgNPs [8,9], with a maximum dissolution percentage of 0.146% of nominal AgNPcitLcys concentration. The release of Ag ions is considered to be the main, but not the only, mechanism of AgNP toxicity towards non-target species. AgNP dissolution can be controlled by selecting the appropriate surface coating molecules, such as those containing reduced sulphur groups, which covalently bind to Ag atoms onto the NP surface, thereby reducing contact with oxidizing agents and consequent Ag dissolution [6,7,8,9].

### 3.2. Ecotoxicity

For the description of morphological alteration and regeneration capacity, we used the evaluation proposed by Wilby [29], by assigning a score from 1 to 10 based on the observed morphology of the specimens. For the morphological assay, the evaluation was performed every 24 h up to 96 h of exposure, while for the regeneration assay, it was performed after 96 h and after 7 days of exposure.

The exposure of *H. vulgaris* to AgNPcitLcys up to 1000 µg/L showed minimal consequences for both morphology and regeneration, with the exception of a slight contraction of column and tentacles at 100 and 1000 µg/L (Figure 1, upper row; Figure 2A). In contrast, a dose-dependent reduction in morphological parameters was observed upon AgNO_3_ exposure, with organisms reaching the tulip shape at the highest concentration tested (100 µg/L) (Figure 1, lower row; Figure 2B). The higher toxicity of AgNO_3_ compared to AgNPs agrees with previous observations, as the ionic form is more available for biological uptake in short-term exposure, while the effects of AgNP depend on particle dissolution and uptake and usually need longer exposure times to be observed. Compared to available studies on AgNPs ecotoxicity on *Hydra* (Kang and Park, 2021; Auclair and Gagné, 2022), AgNPcitLcys exhibited lower toxicity, causing only mild morphological alterations (100–1000 µg/L) and no mortality. In the regeneration assay, both AgNPcitLcys- and AgNO_3_-exposed organisms showed a high regeneration rate, reaching a score of 8 or more after 7 days of exposure, even at the highest concentration tested (Figure 2C,D).

The low acute ecotoxicity of AgNPcitLcys was confirmed for the freshwater cnidarian *H. vulgaris*, as also previously observed for freshwater and marine water microalgae and microcrustaceans [9]. The low release of Ag ions in *Hydra* medium, as shown by ICP-MS data, surely played a significant role in the observed reduced ecotoxicity, also based on the morphological effects observed upon exposure to AgNO_3_, confirming the sensitivity of *Hydra* to Ag. However, Kang and Park [30] measured high levels of Ag inside *Hydra* specimens exposed to sulfide-coated AgNPs (Ag_2_S-NPs), even if the nominal dissolution of AgNPs in the medium was as low as non-detectable (LOD not reported). This was associated with toxicity, both at the morphological and regeneration levels (with EC_50_ of 0.65 and 0.09 mg/L, respectively). The authors hypothesized the ingestion of Ag_2_S-NPs followed by dissolution, triggered by the acidic environment of the gastrovascular cavity [30].

Previous findings showed effects upon AgNPcitLcys exposure despite the low dissolution levels [9]. This led to hypothesizing the occurrence of a nano-size-related toxic effect for AgNPcitLcys, especially when considering freshwater species. This hypothesis was supported by the low aggregation of AgNPcitLcys in freshwater media, which could have eased intracellular uptake and consequent dissolution and toxicity. In the current work, as shown by DLS data, AgNPcitLcys do undergo a moderate aggregation when dispersed in *Hydra* medium (HD_d_ of 676 ± 10 nm), hence making an intracellular uptake less likely to happen. This could explain the almost total absence of toxicity observed for *H. vulgaris* compared to previously tested freshwater species.

Although Ag levels were not measured in the whole body of *Hydra* and the possibility of ingestion or intracellular uptake cannot be ruled out, the low ecotoxicity seems to confirm the protective role of the double coating of citrate and L-cysteine in preventing Ag ion release and thus ecotoxicity [8,9].

Among regenerated *Hydra* from AgNPcitLcys exposure, some appeared to be bending their tentacles towards the mouth, as shown in Figure 3A, a behavior usually observed during feeding. As a consequence of this observation and since no major disruption occurred for regenerated *Hydra*, a feeding assay was performed in order to check whether the regeneration in a polluted environment could have impaired tentacle functionality. The evaluation of the predation ability (Figure 4) showed a reduction in the number of eaten larvae only for *H. vulgaris* exposed to AgNO_3_ at the highest concentration (100 µg/L). Meanwhile, the number of larvae found dead and considered to be the result of a non-successful predatory attempt was higher than in the control group, in all exposed specimens, even if not statistically significant. The exposure of *Hydra* to both heavy metals and NPs was shown to alter the structure of battery cell complexes (BCCs) [31,32,33] and epithelio-muscular cells that house the cnidocytes and associated neurons in the tentacles, which are responsible for its predating behavior. The impairment of BCCs can affect the feeding behavior of *H. vulgaris*, which normally consists of the following steps: catching and killing of the prey by cnidocytes-armed tentacles, tentacle bending towards the mouth, mouth opening, and ingestion of the prey [34]. The falling off of caught preys could be caused by a reduced entangling ability of the tentacles following an alteration of BCCS. For AgNPcitLcys exposure, this was particularly emphasized at the lowest tested concentration (10 µg/L) (Figure 4A). Similarly, the freshwater microcrustacean *Ceriodaphnia dubia* was shown to suffer more from exposure to a lower concentration of AgNPcitLcys (1 µg/L) compared to an intermediate one (10 µg/L) [9]. AgNPs at lower concentrations (µg/L range) are subjected to reduced aggregation but usually undergo enhanced dissolution compared to higher AgNP concentrations (mg/L–g/L range). Moreover, lower aggregation corresponds to an increased number of AgNPs in their nano-size range, potentially able to cross cell membranes and exert a nano-related toxicity, as previously hypothesized for AgNPcitLcys [9]. This interplay of factors is reflected in toxicity results, with a non-linear trend of responses, and could be responsible for the occurrence of stronger toxic effects at lower concentrations.

Furthermore, recent findings [25,26,35,36] have demonstrated that peptides extracted from *Hydra* basal disks, which are involved in the organism’s defense against bacterial attacks, show a natural ability to counteract AgNP toxicity towards different organisms and cells. Further investigations are thus encouraged to better understand the mechanisms behind such an ability for future biotechnological applications.

## 4. Conclusions

Silver nanoparticles for environmental applications, with a double coating of citrate and L-cysteine, were evaluated for their ecotoxicity towards the freshwater cnidarian *H. vulgaris*. The results confirm the low release of Ag ions from AgNPcitLcys and their safety for *H. vulgaris* up to 1000 µg/L, as previously observed for other freshwater and marine invertebrate model species. These results further support the leading role of surface coating in AgNP environmental safety and highlight the need to focus research efforts on the investigation of the main features involved in AgNP ecotoxicity, in order to allow the production of ecologically safe ENMs for environmental applications.

## Figures and Tables

**Figure 1 toxics-10-00695-f001:**
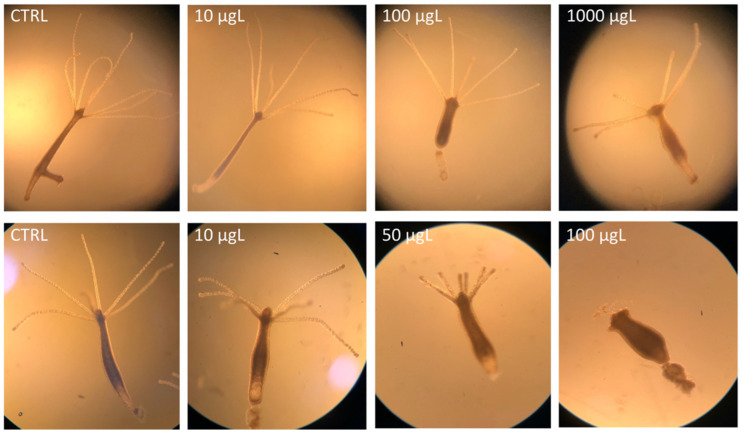
Morphology of *H. vulgaris* exposed to AgNPcitLcys (**upper row**) and AgNO_3_ (**lower row**) for 96 h at 0 (CTRL), 10, 100, and 1000 µg/L and 0 (CTRL), 10, 50, and 100 µg/L, respectively.

**Figure 2 toxics-10-00695-f002:**
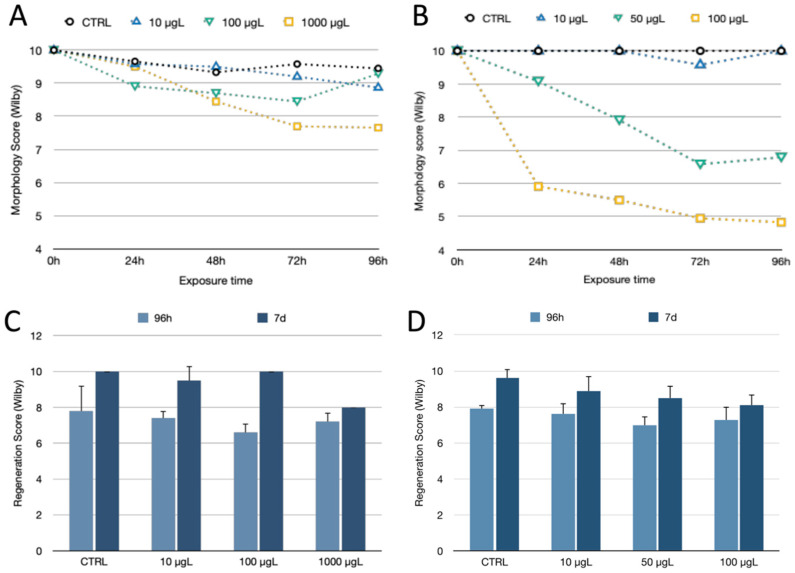
Morphology and regeneration score according to Wilby for *H. vulgaris* exposed to AgNPcitLcys (**A**,**C**) and AgNO_3_ (**B**,**D**) for 96 h (morphology) and 96 h and 7 days (morphology and regeneration) at 0 (CTRL), 10, 100, and 1000 µg/L for AgNPcitLcys and 0 (CTRL), 10, 50, and 100 µg/L for AgNO_3_. Each data point is the mean value obtained from 12 specimens with standard deviation.

**Figure 3 toxics-10-00695-f003:**
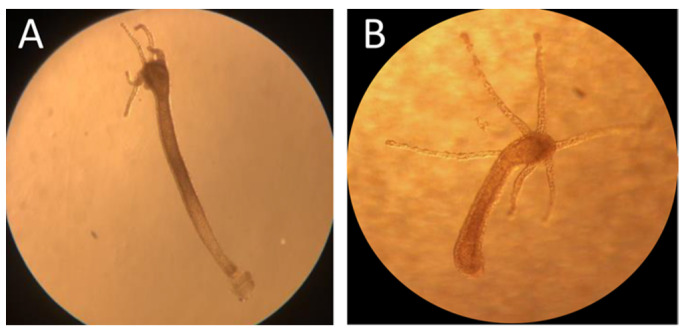
Regeneration of *H. vulgaris* columns exposed to AgNPcitLcys (**A**) and AgNO_3_ (**B**) for 7 days at 1000 µg/L and 100 µg/L, respectively.

**Figure 4 toxics-10-00695-f004:**
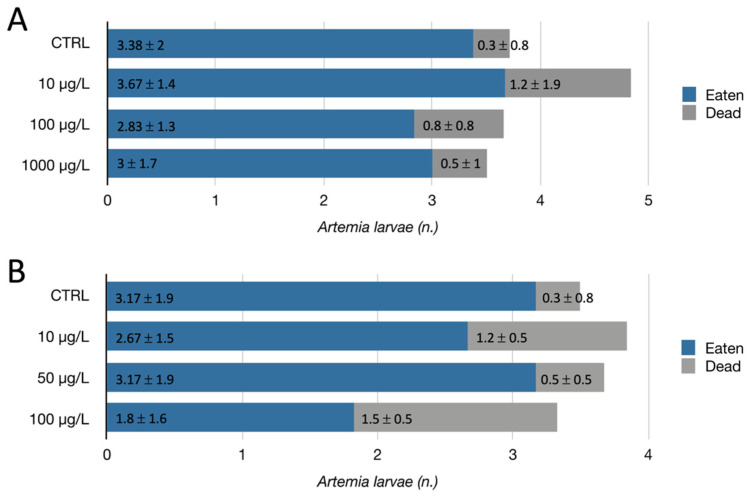
Evaluation of predation ability of *H. vulgaris* after 7 days of exposure to (**A**) AgNPcitLcys and (**B**) AgNO_3_. The graphs show the average number (± standard deviation) of *A. franciscana* larvae either eaten (blue) or found dead at the bottom of the well (gray).

**Table 1 toxics-10-00695-t001:** Average hydrodynamic diameter (HD_d_, reported in intensity and volume), polydispersity index (PDI), and surface charge (ζ-potential) of AgNPcitLcys as measured by DLS in MilliQ water and *Hydra* medium at 50 mg/L and 25 °C. Data are shown as mean ± standard deviation.

	HD_d_ (nm) Intensity	HD_d_ (nm) Volume	PDI	ζ-Potential (mV)
MilliQ	136 ± 11	14 ± 9	0.5	–47.9
*Hydra* medium	676 ± 10	1283 ± 182	0.28	–18.7

**Table 2 toxics-10-00695-t002:** Reported Ag values (µg/L) as measured by ICP-MS in *Hydra* medium and *Hydra* medium with 1000 µg/L AgNPcitLcys after 1 h, 96 h, and 7 days after filtration (3 kDa). Data are shown as mean ± standard deviation and dissolution percentage of nominal AgNPcitLcys.

	*Hydra* Medium	Hydra Medium + AgNPcitLCys 1 h	Hydra Medium + AgNPcitLCys 96 h	Hydra Medium + AgNPcitLCys 7 days
Ag (µg/L)	0.23 ± 0.07	0.41 ± 0.03 (0.041%)	1.42 ± 0.05 (0.142%)	1.69 ± 0.04 (0.169%)

## Data Availability

Not applicable.

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
