# Peer review of "Sensitivity of Hydra vulgaris to Nanosilver for Environmental Applications"

_toxics, 2022, doi:10.3390/toxics10110695_

Round 1
Reviewer 1 Report
Dear Editor, thank you very much for your invitation to review Manuscript ID toxics-1996209, submitted to “Toxics”.
The original Communication, “Sensitivity of Hydra vulgaris to nanosilver for environmental applications” by Arianna Bellingeri et al., provides valuable information regarding the sensibility of freshwater cnidarian to silver nanoparticles and their use such as a model for engineered nanomaterials safety.
Considering that nanomaterials have been increased in the environment, this study intends to fill out the current knowledge gaps and, from this point of view, it has merits for publication. However, I do not recommend publishing the paper at the current stage because the authors need to make a few improvements:
Abstract: This section is well described and explained. I can find all the necessary elements to build an excellent abstract, like Contextualization, Gap, Purpose, Methods, Results, and Conclusions.
- I would say that the authors should include the number of individuals (n = ?) here: “H. vulgaris were evaluated for morphological aberration after 96h exposure and regeneration ability after 96h and 7d exposure, after which also the predatory ability was assessed.”.
Introduction: I suggest that the authors review the introduction. Regarding the contextualization | state of the art, gap, and purpose: Why is this area important? What has been done before? What still needs to be done? Why is this study important? What is presented here? All this information could be more precise throughout the section.
Materials and Methods: This section is detailed.
- The graph presented the results as mean ± standard error. Is it right?
Results, Discussion and Conclusions: These sections are well summarized.
References:
- Good references. There are some mistakes that the authors need to fix based on the journal’s suggested format.
* After these improvements, the manuscript will be suitable for publication.
Author Response
Reviewer_1
Dear Editor, thank you very much for your invitation to review Manuscript ID toxics-1996209, submitted to “Toxics”.
The original Communication, “Sensitivity of Hydra vulgaris to nanosilver for environmental applications” by Arianna Bellingeri et al., provides valuable information regarding the sensibility of freshwater cnidarian to silver nanoparticles and their use such as a model for engineered nanomaterials safety.
Considering that nanomaterials have been increased in the environment, this study intends to fill out the current knowledge gaps and, from this point of view, it has merits for publication. However, I do not recommend publishing the paper at the current stage because the authors need to make a few improvements:
Abstract: This section is well described and explained. I can find all the necessary elements to build an excellent abstract, like Contextualization, Gap, Purpose, Methods, Results, and Conclusions.
- I would say that the authors should include the number of individuals (n = ?) here: “H. vulgaris were evaluated for morphological aberration after 96h exposure and regeneration ability after 96h and 7d exposure, after which also the predatory ability was assessed.”.
We thank the reviewer for the suggestion, the number of individuals for each exposure concentration was added in the abstract as it follows:
“H. vulgaris (n= 12) were evaluated for morphological aberration after 96h exposure and regeneration ability after 96h and 7d exposure, after which also the predatory ability was assessed”.
Introduction: I suggest that the authors review the introduction. Regarding the contextualization | state of the art, gap, and purpose: Why is this area important? What has been done before? What still needs to be done? Why is this study important? What is presented here? All this information could be more precise throughout the section.
We thank the reviewer for the valuable suggestion, the introduction has been revised following the logical scheme suggested.
Materials and Methods: This section is detailed.
- The graph presented the results as mean ± standard error. Is it right?
That is correct, thank for the observation. The graph captions were changed in order to clarify what is shown.
Results, Discussion and Conclusions: These sections are well summarized.
References:
- Good references. There are some mistakes that the authors need to fix based on the journal’s suggested format.
We thank the reviewer for the comment, the references were revised following the journal guidelines.
* After these improvements, the manuscript will be suitable for publication

Reviewer 2 Report
1. The progress of the relevant studies should be stated in the Introduction.
2. The characterization results of nanosilver can be added to supplementary material.
3. Since the authors have merged the Results and the Discussion, there should be a responsive discussion after every phenomenon description, such as the Section 3.1.
4. Where is the Section 3.2?
5. There are serious problems with the format of the figures, and the authors should make the modification.
Author Response
Reviewer_2:
- The progress of the relevant studies should be stated in the Introduction.
We thank the reviewer for the comment, the introduction has been revised accordingly.
- The characterization results of nanosilver can be added to supplementary material.
We appreciate the comment, but we disagree with the reviewer in moving the characterization of the AgNPs to supplementary material since AgNP behavior and properties in exposure media are mandatory to understand the exposure conditions and the potential mechanisms of action. Please refer to the new guideline of OECD on aquatic and sediment ecotoxicological testing of nanomaterials: https://www.oecd.org/officialdocuments/publicdisplaydocumentpdf/?cote=env/jm/mono(2020)8&doclanguage=en
- Since the authors have merged the Results and the Discussion, there should be a responsive discussion after every phenomenon description, such as the Section 3.1.
We thank the reviewer for the valuable suggestion, The result and discussion section was revised in order to better discuss every result.
- Where is the Section 3.2?
Section 3.3 was misnumbered, there are only two sections: 3.1 and 3.2.
- There are serious problems with the format of the figures, and the authors should make the modification.
The conversion of the manuscript into pdf causes problems with the figures format, this does not happen in the word version in which the figures are perfectly visible.

Reviewer 3 Report
There are not information regarding the pH induced by the Nanosilver particles with citrate and L-cysteine capping the are added in to the medium. On the other hand there are no informations regarding the pH supported by Hydra vulgaris in environmental medium. Please mention in the the methodology the both values for the pH.
The study added novelty in the field of eco-toxicity. I recommend the publication after the corrections will be made it.
Author Response
Reviewer 3:
There are not information regarding the pH induced by the Nanosilver particles with citrate and L-cysteine capping the are added in to the medium. On the other hand there are no informations regarding the pH supported by Hydra vulgaris in environmental medium. Please mention in the the methodology the both values for the pH.
We thank the reviewer for the comment, this information was not mentioned in the text but was previously checked and showed that Hydra medium present a physiological pH around 8 and the addition of AgNPcitLcys suspension does not change this value.
A sentence was added in the materials and method section in the text as it follows: “During the test we ensured that pH values of exposure solutions stayed in the physiological range of 7.8-8, as that of Hydra medium with no AgNPcitLcys. This was also facilitated by the fact that AgNPcitLcys were suspended in water with no added surfactants or additives and that the volume of AgNPcitLcys stock solutions added in each exposure well did not exceed 20 µL.”
The study added novelty in the field of eco-toxicity. I recommend the publication after the corrections will be made it.

Round 2
Reviewer 2 Report
It can be published in current version.